# The Diversity of *Wolbachia* and Other Bacterial Symbionts in *Spodoptera frugiperda*

**DOI:** 10.3390/insects15040217

**Published:** 2024-03-22

**Authors:** Yuan Liu, Lina Zhang, Xiangyun Cai, Alexandre Rutikanga, Baoli Qiu, Youming Hou

**Affiliations:** 1State Key Laboratory of Ecological Pest Control for Fujian and Taiwan Crops, Ministerial and Provincial Joint Innovation Centre for Safety Production of Cross-Strait Crops, Fujian Agriculture and Forestry University, Fuzhou 350002, China; yuanliu78@163.com (Y.L.);; 2Guangdong Laboratory for Lingnan Modern Agriculture, Guangzhou 510642, China; 3Engineering Research Center of Biotechnology for Active Substances, Ministry of Education, Chongqing Normal University, Chongqing 401331, China; 4College of Agriculture and Animal Husbandry, University of Rwanda, Kigali 999051, Rwanda

**Keywords:** fall armyworm, 16S rRNA high-throughput sequencing, bacterial symbionts, *w*Fru strain, infection rates

## Abstract

**Simple Summary:**

Bacterial symbionts, especially *Wolbachia*, are vital in many physiological processes of insects. However, the mean infection prevalence of *Wolbachia* in many species of lepidopteran insects is relatively low. Here, we investigated the infection, composition, abundance, and diversity of bacterial symbionts, especially *Wolbachia*, associated with *Spodoptera frugiperda.* Our results revealed that *Wolbachia* was found in the ovaries and salivary glands of *S. frugiperda* female adults. Although the infection and abundance of *Wolbachia* varied between geographical populations, they all belonged to the supergroup B and were named the *w*Fru strain, which has been considered to potentially induce cytoplasmic incompatibility. These findings may provide a foundation for developing potential biocontrol techniques for *S. frugiperda*.

**Abstract:**

Bacterial symbionts associated with insects can be crucial in insect nutrition, metabolism, immune responses, development, and reproduction. However, the bacterial symbionts of the fall armyworm *Spodoptera frugiperda* remain unclear. *S. frugiperda* is an invasive polyphagous pest that severely damages many crops, particularly maize and wheat. Here, we investigated the infection, composition, abundance, and diversity of bacterial symbionts, especially *Wolbachia*, in different tissues of *S. frugiperda* female adults. The infection prevalence frequencies of *Wolbachia* in five provinces of China, namely Pu’er, Yunnan; Nanning, Guangxi; Sanya, Hainan; Yunfu, Guangdong; and Nanping, Fujian, were assessed. The results indicated that Proteobacteria, Firmicutes, and Bacteroidetes were the three most dominant bacterial phyla in *S. frugiperda* adults. At the genus level, the abundant microbiota, which included *Enterobacter* and *Enterococcus*, varied in abundance between tissues of *S. frugiperda*. *Wolbachia* was found in the ovaries and salivary glands of *S. frugiperda* adults, and was present in 33.33% of the Pu’er, Yunnan, 23.33% of the Nanning, Guangxi, and 13.33% of the Sanya, Hainan populations, but *Wolbachia* was absent in the Yunfu, Guangdong and Nanping, Fujian populations. Further phylogenetic analyses revealed that all of the *Wolbachia* strains from the different *S. frugiperda* populations belonged to the supergroup B and were named the *w*Fru strain. Since there were *Wolbachia* strains inducing cytoplasmic incompatibility in supergroup B, these findings may provide a foundation for developing potential biocontrol techniques against *S. frugiperda*.

## 1. Introduction

The fall armyworm *Spodoptera frugiperda* (J.E. Smith) (Lepidoptera: Noctuidae), which is native to tropical and subtropical areas in America [1,2], is considered one of the most serious, polyphagous, invasive insect pests in the world [3]. *S. frugiperda* displays two subpopulations called the “Rice” and “Corn” strains, which have affected more than 350 species of plants from 76 families, including maize, wheat, rice, cotton, sugarcane, and sorghum [4]. In China, *S. frugiperda* was first found in western Yunnan Province in 2018 [5] and then rapidly expanded to the other 26 provinces within a year, except for Heilongjiang, Jilin, Liaoning, Qinghai, and Xinjiang [6,7,8]. *S. frugiperda* has evolved to acquire strong resistance against most insecticides (e.g., pyrethroids, organophosphates, and carbamates) [9] and overcome the resistance of transgenic maize [10,11]. Consequently, alternative control strategies for *S. frugiperda* management are urgently needed, and population manipulation with bacterial symbionts is one of the emerging strategies used in pest management [12,13,14].

Biological interactions between bacteria and insects, which can be parasitic, symbiotic, or neutral, are very common in nature [15]. Insects generally host a gut microbiota, which is extracellular and can vary depending on the environmental conditions and on the insect’s diet, and endosymbiotic bacteria, which are intracellular and can be obligate or facultative [16]. Previous studies have revealed that symbiotic bacteria play vital roles in many physiological processes of insects, including nutrition, metabolism, immune responses, development, and reproduction [17,18]. For instance, the IIa bacteriocin (mundticin KS) produced by the gut bacterium *Enterococcus mundtii* inhibited pathogen colonization in *Spodoptera littoralis* (Lepidoptera: Noctuidae) [19]. Xia et al. [20] revealed that the most abundant gut bacteria (*Enterobacter* spp.) contributed to plant cell wall degradation in *Plutella xylostella* (Lepidoptera: Plutellidae) and enhanced food utilization efficiency. Wang et al. [21] hypothesized that *Wolbachia* may be a crucial factor influencing the reproductive isolation of *Ectropis obliqua* (Lepidoptera: Geometridae) and *Ectropis grisescens* (Lepidoptera: Geometridae). In addition, Lei et al. [22] reported that *Wolbachia* reduced the susceptibility of the striped stem borer *Chilo suppressalis* (Lepidoptera: Crambidae) to two insecticides (fipronil and avermectin). Among the intracellular bacteria endosymbionts, *Wolbachia* (Alphaproteobacteria: Rickettsiales) is one of the most abundant in nematodes, insects, and other arthropods, and approximately 40–60% of all insect species in nature are estimated to be infected with *Wolbachia* [23,24]. Normally, these bacteria can manipulate the reproductive pattern of their hosts in diverse ways, including through cytoplasmic incompatibility (CI), feminization, male killing (MK), and parthenogenesis (PI), to increase the chance of infected individuals reproducing successfully and spreading the infection through their progeny [25,26]. *Wolbachia*, due to its inability to be cultured outside host cells, and the infection rate of *Wolbachia* in many species of lepidopteran insects is relatively low, which presents a serious challenge for studying its infection patterns [27,28,29]. Consequently, understanding the composition, diversity, and potential functions of bacterial symbionts in *S. frugiperda*, particularly regarding *Wolbachia*, is a gray area in research.

In the present study, 16S rRNA high-throughput sequencing and polymerase chain reaction (PCR) were used to investigate the infection, composition, abundance, and diversity of bacterial symbionts in *S. frugiperda*, with a particular focus on *Wolbachia*. Given the recognized potential of *Wolbachia* for biological control and the paucity of research on its interaction with *S. frugiperda*, our study of pest-associated symbiotic bacteria, especially *Wolbachia*, may lay a useful foundation for developing potential biocontrol techniques against *S. frugiperda*.

## 2. Materials and Methods

### 2.1. Insect Rearing and Plant Growth

*Spodoptera frugiperda* were initially collected from a corn field in July 2019 in Zhangzhou (24.52° N, 117.35° E), Fujian Province, Southeast China. *S. frugiperda* larvae were reared on fresh maize leaves in a lattice box (inner grid size: 4.1 cm, × 2.3 cm, × 3.3 cm, 3 × 8 grids) under a 14:10 h light–dark cycle in an incubator at 26 ± 1 °C and 70 ± 10% relative humidity (RH). Thereafter, the pupae were transferred to a new cage (35 cm, × 35 cm, × 35 cm), where emerging adults were fed an artificial diet with a 10% honey solution under the same conditions for reproduction. The experimental populations were reared for at least 10 generations before use.

To supply maize leaves as food for *S. frugiperda*, seeds (*Zea mays* L. var. Zhengdan no. 958) were sown in soil (10% sand, 5% clay, and 85% peat) in a greenhouse under ambient conditions. Plants without any pest or disease damage were used at the 6–12-expanded leaf stage.

### 2.2. PCR Amplification, Library Preparation, and Sequencing

Approximately 30 newly emerged, unmated *S. frugiperda* female adults were sterilized with 75% alcohol and dissected under a stereomicroscope on an ultraclean workbench, after which gut, ovary, salivary gland, and fat body tissues were isolated. These tissues were divided into three replicates for testing. Total genomic DNA for all samples was extracted using the TIANamp Genomic DNA Kit (Tiangen, Beijing, China) following the manufacturer’s instructions. DNA extraction was also carried out on an ultraclean workbench after ultraviolet sterilization to avoid contamination by environmental DNA. The specific primers 27F (5′-AGRGTTTGATYNTGGCTCAG-3′) and 1492R (5′-TASGGHTACCTTGTTASGACTT-3′) were used to amplify the V1–V9 regions of the *16S rRNA* gene. PCR amplification was performed using the following thermocycling procedures: 95 °C for 5 min, 25 cycles of 95 °C for 30 s, 50 °C for 30 s, and 72 °C for 40 s, with a final elongation step at 72 °C for 7 min. Homogenized PCR products were visualized through 1.5% agarose gel electrophoresis, with gel purified using a TIANgel Midi Purification Kit (Tiangen, Beijing, China), and subsequently homogenized to generate a sequencing library (SMRT Bell). The quality of the established library was checked first, and single-molecule sequencing was conducted on the PacBio Sequel platform at Biomarker Bioinformatics Technology, Co., Ltd. (Beijing, China). Filtering and demultiplexing of the raw readings were performed to obtain circular consensus sequence (CCS) readings via SMRT Link software (version 8.0) (minPasses ≥ 5 and minPredictedAccuracy ≥ 0.9). Subsequently, the CCS readings were assigned to corresponding samples based on their barcodes using lima v1.7.0. The Cutadapt quality control program (version 2.7) was utilized to eliminate CCS readings lacking primers or falling outside the specified length range (1200–1650 bp). The chimeric sequences were identified and excised employing UCHIME v8.1 [30]. The CCS readings exhibiting a similarity level of 97% or greater were clustered as operational taxonomic units (OTUs) using USEARCH v10.0 [31]. OTUs with sequences < 0.005% of the total number of sequences were discarded [32]. The resulting representative sequences from each OTU cluster were screened for further annotation. Taxonomic classification of all OTUs was conducted using the RDP classifier v2.2, with a confidence threshold of 80% [33], based on the SILVA database (release 132) [34]. Finally, to facilitate downstream diversity analysis, selected OTUs were aligned to the core alignment template of the SILVA database using PyNAST v1.2.2.

### 2.3. Microbiome Diversity Analysis

For further bacterial community analysis, the microbial diversity of each sample was estimated using alpha diversity indices (including the ACE, Chao1, Simpson, and Shannon indices). The ACE and Chao 1 indices are primarily used to estimate species richness, while the Simpson index focuses on dominance, and the Shannon index combines species richness and evenness. These indices, along with the OTU numbers and coverage of the library, were calculated in Mothur v.1.30 [35]. Beta diversity was assessed using QIIME software (version 2), and the microbial community structures across different groups were compared using principal component analysis (PCA) based on the Bray–Curtis index and weighted UniFrac distance matrices [36]. To assess the statistical significance of the differences in the bacterial symbionts among the groups, analysis of similarities (ANOSIM) and permutational multivariate analysis of variance (PERMANOVA) were conducted using the *anosim* and *adonis* functions of the package vegan v2.3.0 [37] in the R v3.1.1 programming environment [38]. In ANOSIM, an *R* value closer to 1 implies greater dissimilarity between groups than within groups [39], while a larger *R*^2^ value in PERMANOVA indicates that the grouping factor significantly contributes to the overall variation [40]. *p* values less than 0.05 in both the ANOSIM and PERMANOVA were considered to indicate statistical significance.

### 2.4. PCR Detection of Endosymbionts from Different S. frugiperda Populations

*S. frugiperda* female adults were collected from five collection localities in China, namely Pu’er, Yunnan; Nanning, Guangxi; Sanya, Hainan; Yunfu, Guangdong; and Nanping, Fujian (Figure 1). Approximately 30 adults from each province were immersed separately in 95% ethyl alcohol and frozen at −20 °C before being assessed for the infection prevalence frequencies of endosymbionts, including *Wolbachia*, *Arsenophonus*, *Cardinium*, *Rickettsia*, and *Spiroplasma*, which have been reported to be present in many species of studied lepidopteran insects [41]. *Enterobacter asburiae*, which exhibited 100% infection among the samples analyzed, was used as a positive control. The sample information is provided in Table 1.

Total DNA was extracted from a single *S. frugiperda* female adult following the method of Ahmed et al. [42], where 30 individuals were tested in each province, 10 individuals were treated as one experimental replicate to determine the infection prevalence frequencies of endosymbionts, and three replicates were performed. PCR amplification, which included assays for *16S rRNA* (targeting conserved regions within the ribosomal RNA of *Wolbachia*), *wsp* (encoding the *Wolbachia* surface protein), and *16S rDNA* (amplifying a fragment that includes the *16S rRNA* coding sequence and other ribosomal DNA elements of *Wolbachia*) genes of endosymbionts, was performed in 25 μL of buffer containing 1 μL of the template DNA lysate, 1 μL of each primer, 2.5 mM MgCl_2_, each dNTP at 200 mM, and 1 unit of DNA Taq polymerase (Invitrogen, Guangzhou, China) [43]. The specific primers used in this study are shown in Table 2. The remainder of the PCR products of single colonies were then sent to the Beijing Institute (BGI) for sequencing after the expected bands were visible on a 1.5% agarose gel containing Gold-View colorant. The results were compared to known sequences using NCBI’s BLAST algorithm (https://blast.ncbi.nlm.nih.gov/Blast.cgi, accessed on 15 March 2022). Finally, both the species and prevalence frequencies of endosymbionts in the five collection localities were determined.

### 2.5. Phylogenetic Analysis of Wolbachia in S. frugiperda

A phylogenetic tree of *Wolbachia* was constructed based on the *wsp* gene sequence. The *wsp* gene sequences of *Wolbachia* from different insect hosts were selected as references for homology analysis in the GenBank database using basic local alignment search tools (BLASTs). These *wsp* gene sequences of *Wolbachia* are listed in Appendix A, and were edited and aligned using Lasergene v7.1 (DNASTAR, Inc., Madison, WI, USA). The Bayesian information criterion was used to select the best model and partitioning scheme in PartitionFinder v.1.0.1 [51]. In addition, the phylogenetic tree of *Wolbachia* was generated with IQ-TREE v1.6.8 using the TUM+F+R3 model for the *wsp* gene. The phylogenetic tree of *Wolbachia* was constructed based on the maximum likelihood (ML) method with 1000 nonparametric bootstrap replications in RAxML [52].

### 2.6. Statistical Analyses

All of the alpha diversity indices (ACE, Chao1, Simpson, and Shannon indices), circular consensus sequencing (CCS) readings of *Wolbachia* in different tissues, and infection and prevalence of endosymbionts in different *S. frugiperda* populations were analyzed using one-way analysis of variance (ANOVA), and the means were compared using Tukey’s HSD test (SPSS 21.0) at *p* < 0.05. All graphs were drawn with SigmaPlot 10.0.

## 3. Results

### 3.1. Bacterial Abundance in Different Tissues of S. frugiperda

A total of 80,521 CCS readings were obtained after 12 samples were sequenced and identified by barcode; each sample generated at least 4853 CCS readings with an average of 6710 CCS readings. The raw sequence data obtained from this study were deposited in the National Center for Biotechnology Information (NCBI) under accession number PRJNA995535. The clean CCS readings (gut: 6892.67 ± 473.32; ovary: 5588.67 ± 398.40; salivary gland: 6981.00 ± 249.03; fat body: 7375.00 ± 83.74) were retained after primer removal and length filtration. The effective CCS readings (gut: 6881.00 ± 472.02; ovary: 5528.00 ± 427.79; salivary gland: 6828.33 ± 270.84; fat body: 7274.33 ± 71.42) were retained after chimeric read removal. All of the effective tags were clustered into OTUs (gut: 110.00 ± 35.38; ovary: 194.33 ± 33.11; salivary gland: 209.33 ± 18.80; fat body: 44.33 ± 6.69). The Good’s coverage rates were >0.99, indicating that a high degree of sequencing coverage, with all microbiota in each group, was represented by the number of OTUs identified (Table 3).

Based on their abundance and annotation information, 481 bacterial OTUs were assigned to 21 phyla, 33 classes, 81 orders, 148 families, and 290 genera. At the phylum level, the most abundant bacterial phyla identified were Proteobacteria, Firmicutes, and Bacteroidetes, among which Proteobacteria was the most abundant in the fat body tissues of *S. frugiperda* female adults (98.70%). The phyla detected in the ovaries were evenly represented, with Proteobacteria accounting for 65.76%, Firmicutes accounting for 23.40%, and Bacteroidetes accounting for 7.03% (Figure 2A). At the genus level, the abundances of microbiota members *Enterobacter* and *Enterococcus* varied between the tissues of *S. frugiperda*. The abundance of *Enterobacter* in the ovaries (57.07%) was much greater than that in the other three tissues (gut: 44.10%; salivary gland: 3.99%; fat body: 1.21%), while *Enterococcus* was the most abundant genus in the gut tissues (42.83%), representing 6.88% of the ovary, 2.32% of the salivary gland, and 0.42% of the fat body tissues (Figure 2B).

### 3.2. Bacterial Diversity and Community Structure in Different Tissues of S. frugiperda

Microbiome diversity was estimated in relation to the alpha diversity, which was measured as the species richness and diversity in a sample. The Shannon and Simpson diversity metrics demonstrated that the salivary glands had significantly greater microbial diversity than the other tissues (Shannon F_3,8_ = 17.977, *p* = 0.001; Simpson F_3,8_ = 7.443, *p* < 0.05). Similar significant differences were also found for other alpha diversity indices (i.e., ACE F_3,8_ = 4.429, *p* < 0.05; Chao1 F_3,8_ = 3.610, *p* > 0.05) (Figure 3A–D). PCA revealed distinct clustering of microbes from different tissues of *S. frugiperda*, with obvious differences between the four sample groups, including the gut, ovary, salivary gland, and fat body tissues (Figure 3E). Moreover, ANOSIM and PERMANOVA demonstrated significant differences in bacterial communities across samples from the four distinct groups (Table 4).

### 3.3. Abundance of Wolbachia in S. frugiperda

The *Wolbachia 16S rRNA* gene was successfully amplified from *S. frugiperda*, and the sequence was submitted to GenBank (accession number OR268559). 16S rRNA high-throughput sequencing revealed that *Wolbachia* was found in the ovaries and salivary glands of *S. frugiperda* female adults (F_3,8_ = 21.833, *p* < 0.001) (Figure 4).

### 3.4. Infection and Prevalence of Endosymbionts in S. frugiperda Populations

PCR revealed that two endosymbionts, *Wolbachia* (*wsp* gene) and *Arsenophonus* (*16S rRNA* gene), were present in *S. frugiperda*, while *Rickettsia* (*16S rRNA* gene), *Cardinium* (*16S rRNA* gene), and *Spiroplasma* (*16S rRNA* gene) were absent in *S. frugiperda* (Figure 5A). The infection and prevalence of the endosymbionts varied between the five provinces of Southeast China (including Pu’er, Yunnan; Nanning, Guangxi; Sanya, Hainan; Yunfu, Guangdong; and Nanping, Fujian). *Arsenophonus* was detected in only the Nanning, Guangxi populations (F_5,12_ = 410.900, *p* < 0.001), and *Wolbachia* was present in the Pu’er, Yunnan; Nanning, Guangxi; and Sanya, Hainan populations (F_5,12_ = 880.000, *p* < 0.001; F_5,12_ = 410.900, *p* < 0.001; F_5,12_ = 868.000, *p* < 0.001), but it was not detected in the Yunfu, Guangdong and Nanping, Fujian populations (Figure 5B–F).

### 3.5. Wolbachia Infection Rates in S. frugiperda Populations

The *Wolbachia* infection rates varied between the *S. frugiperda* populations, being approximately 33.33% in the Pu’er, Yunnan population; 23.33% in the Nanning, Guangxi population; and 13.33% in the Sanya, Hainan population (F_4,10_ = 32.000, *p* < 0.001) (Figure 5G).

### 3.6. Phylogenetic Analysis of Wolbachia in S. frugiperda Populations

The *wsp* sequences from different *S. frugiperda* populations were deposited in the GenBank database (accession numbers OR282569–OR282571). Phylogenetic analyses of *Wolbachia* based on the *wsp* gene indicated that the tested *Wolbachia* strains from three *S. frugiperda* populations belonged to the supergroup B and were named the *w*Fru strain. In addition, *S. frugiperda* from the Pu’er, Yunnan and Nanning, Guangxi populations were first clustered into one branch, and then clustered with the Sanya, Hainan population to form one peripheral branch (Figure 6).

## 4. Discussion

Insects harbor many symbiotic bacteria, and these bacteria and their hosts have formed complex symbiotic relationships via coevolution [53,54]. Growing evidence has revealed that manipulating the bacterial communities of insects could be a feasible strategy for reducing the incidence of improper and excessive application of insecticides [55,56]. Thus, in this study, 16S rRNA high-throughput sequencing and polymerase chain reaction (PCR) methods were used to determine the composition, diversity, and potential function of bacterial symbionts, especially *Wolbachia*, of *S. frugiperda*.

Lepidopteran insects comprise the second most diverse insect order, with some of the most devastating agricultural pests worldwide [57]. Recently, extensive evidence has shown that developmental stage, diet, host phylogeny, and habitat environmental conditions determine insect bacterial diversity and community structure [17]. However, the composition of the bacterial microbiota is relatively simple in lepidopteran insects, including *S. frugiperda* [58,59,60]. At the phylum level, Xia et al. [20] reported that Proteobacteria was the dominant taxon (from 70.42% to 97.44% in larvae, 97.44% to 99.58% in pupae, and 99.64% to 100% in adults), followed by Firmicutes (from 2.45% to 29.48% in larvae, 0.42% to 1% in pupae, and 0 to 0.36% in adults), in the *P. xylostella* gut. Dantur et al. [61] suggested that the bacterial genes of Proteobacteria and Firmicutes can degrade cellulose, hemicellulose, and pectin in *Spodoptera litura* (Lepidoptera: Noctuidae). Similarly, our study demonstrates that the bacterial communities in different female tissues of *S. frugiperda* mostly comprised Proteobacteria, Firmicutes, and Bacteroidetes, which are considered constituents of the core microbiota. These findings are consistent with those of some studies describing the microbiota in other lepidopteran insects, such as *Lymantria dispar*, *Helicoverpa armigera*, *Manduca sexta*, *Bombyx mori*, and *Spodoptera exigua* [62,63,64,65,66].

In general, at the genus level, the predominant microbiota members include *Enterobacter*, *Enterococcus*, *Pseudomonas*, *Bacillus*, and *Staphylococcus*, which are present in more than 70% of the studied lepidopteran insects [17,20,66]. Among these bacteria, *Enterococcus* and *Enterobacter* have been implicated in the degradation of the plant cell wall, which can release available nutrients in *P. xylostella* [67]. In addition, a high abundance of *Enterobacter* can accelerate the development of resistance to insecticides in *P. xylostella* [68]. In previous studies, *Enterococcus* was found to diminish the insecticidal efficacy of *Bacillus thuringiensis* (Bt) in *Lymantria dispar* (Lepidoptera: Liparidae) larvae [69]. Similarly, the gut microbiota of *Manduca sexta* (Lepidoptera: Sphingidae) and *P. xylostella* was reported to reduce the effectiveness of the Cry1Ac toxin [70]. In line with these findings, our results also showed that the dominant bacterial genera were *Enterobacter* and *Enterococcus* in different tissues, which may offer new insights into the chemical resistance mechanism of *S. frugiperda* in the future. Alpha and beta diversity analyses revealed significant differences in microbes from different tissues of *S. frugiperda*, and these findings underscore the complexity of microbial ecosystems. In addition, growing evidence has shown that a range of ecological interactions, evolutionary histories, and environmental factors may contribute to differences in insect diversity [71,72].

16S rRNA high-throughput sequencing suggested that the intracellular bacteria endosymbiont *Wolbachia* was found in the ovaries and salivary glands of *S. frugiperda* female adults. Moreover, our results are similar to the findings of a previous study by Ou et al. [73], in which *Wolbachia* was detected in the midgut, salivary glands, testes, and ovaries of *Diaphorina citri* (Hemiptera: Psyllidae) and *Cornegenapsylla sinica* (Hemiptera: Psyllidae). To date, approximately 80% of lepidopteran insects are estimated to be infected by *Wolbachia* [74]. However, the mean infection prevalence (proportion of infection within a population) in many species of lepidoptera is relatively low, at approximately 27% [17]. Some of these studies have revealed that *Wolbachia* infection rates may be related to geographical distribution [29,41,75]. For instance, Wang et al. [21] reported that *Wolbachia* infection rates differed between all tested *E. grisescens* samples collected from eight geographical tea-producing areas (Xinchang, Yuhang, Liyang, Langxi, Nanchang, Enshi, Guiyang, and Guiyang) in China. Zhu et al. [76] reported that 7% of *P. xylostella* was infected with *Wolbachia*, but only in individuals from South America, North America, Africa, and Asia, as no infections were detected in populations from Europe and Oceania. Furthermore, different environmental factors strongly influence the *Wolbachia* titer [77,78]. In the butterfly *Zizeeria maha* (Lepidoptera: Lycaenidae), the *Wolbachia* density decreased across several host generations from spring/early summer to autumn [79]. In the present study, we showed that *Wolbachia* was present in only Pu’er, Yunnan; Nanning, Guangxi; and Sanya, Hainan, with infection rates of 33.33%, 23.33%, and 13.33%, respectively, and no infection was detected in populations from Yunfu, Guangdong and Nanping, Fujian. This difference may be correlated to the host’s genetic variation, diet, temperature, and other abiotic factors [80].

*Wolbachia* strains are highly diverse among lepidopteran insects, consisting of 90 different strains [81]. For example, Zhu et al. [76] reported three *Wolbachia* strains infecting *P. xylostella*, with *plutWA1* and *plutWA2* belonging to supergroup A and *plutWB1* belonging to supergroup B. Sakamoto et al. [82] reported that infection with the *Wolbachia* strain *w*Fur in *Ostrinia scapulalis* (Lepidoptera: Crambidae) caused feminization of genetic males. In addition, molecular typing analyses demonstrated that these *Wolbachia* strains in all of the tested *E. grisescens* samples were of the same *w*Gri strain, and could strongly induce CI and enhance the fecundity of its female hosts [18]. In our study, phylogenetic analyses based on the *wsp* gene revealed that all of the *Wolbachia* strains from different *S. frugiperda* populations belonged to the supergroup B and were named the *w*Fru strain. Since these were the *Wolbachia* strains inducing CI in supergroup B [83], this possibility may also occur in the *Wolbachia* strain *w*Fru of *S. frugiperda*. Moreover, *Wolbachia* can also help the host synthesize molecules, which is essential for growth and development, alter host microbial communities, and protect hosts against a variety of pathogens and abiotic stresses [84,85,86]; thus, the functions of the *Wolbachia* strain *w*Fru in *S. frugiperda* are worthy of further investigation.

In conclusion, our results revealed that Proteobacteria, Firmicutes, and Bacteroidetes were the three most dominant bacterial phyla in different tissues of *S. frugiperda* female adults. The dominant bacterial genera were *Enterobacter* and *Enterococcus*, which varied in abundance between tissues. In addition, *Wolbachia* was found in the ovaries and salivary glands of *S. frugiperda* female adults. Although the infection and abundance of *Wolbachia* varied among different geographical populations, they all belonged to the supergroup B and were named the *w*Fru strain, which has been considered to potentially induce cytoplasmic incompatibility. Further studies are needed to elucidate the host phenotypes of *Wolbachia* infection and its usefulness as a biological control agent for *S. frugiperda* management.

## Figures and Tables

**Figure 1 insects-15-00217-f001:**
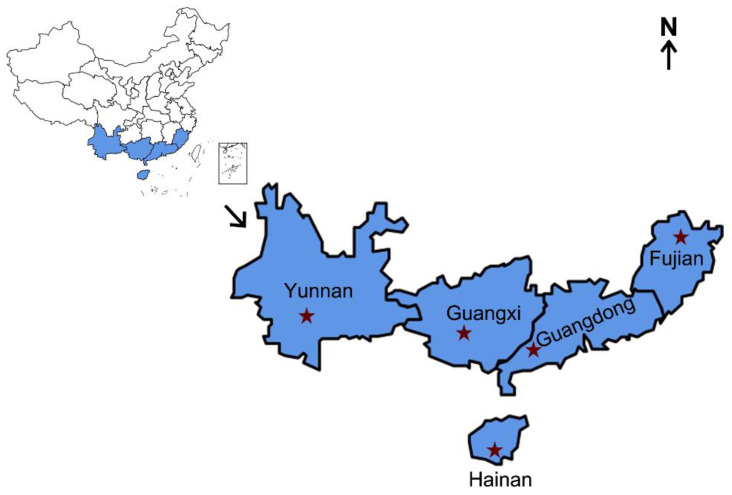
Collection regions of *S. frugiperda*. The blue-shadowed parts represent the sampled provinces, and the red stars represent the exact locations. Information on the different *S. frugiperda* samples is provided in Table 1.

**Figure 2 insects-15-00217-f002:**
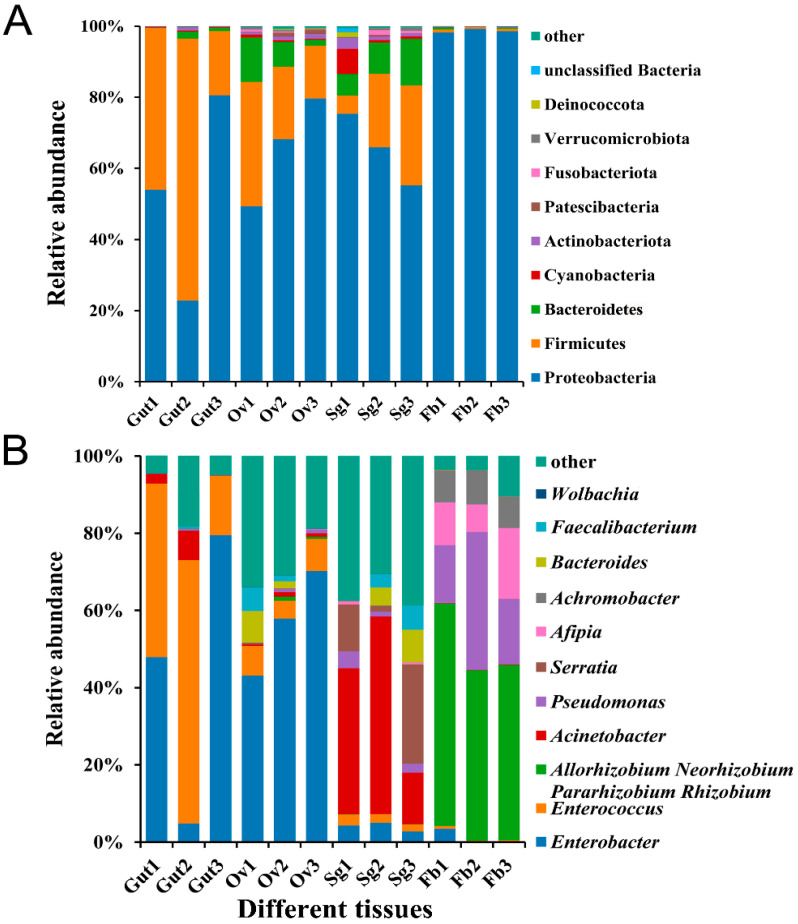
Relative bacterial abundance in different tissues of *S. frugiperda* female adults. (**A**) Phylum-level relative abundances of bacteria in *S. frugiperda*; (**B**) genus-level relative abundances of bacteria in *S. frugiperda*.

**Figure 3 insects-15-00217-f003:**
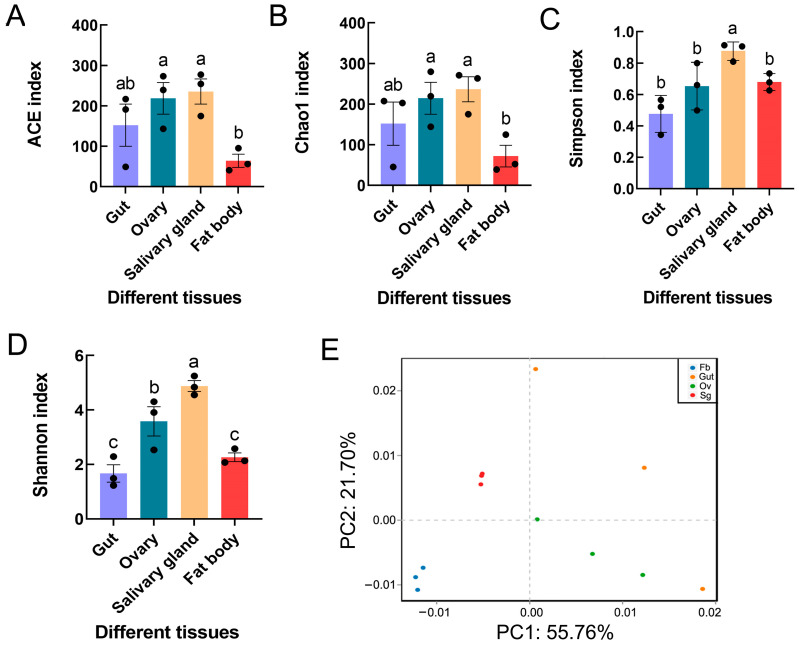
Bacterial diversity and community structure in different tissues of *S. frugiperda* female adults: (**A**) ACE index; (**B**) Chao1 index; (**C**) Simpson index; (**D**) Shannon index. The columns and error bars represent the means ± SEs (n = 3). The different letters above the bars indicate significant differences between the tissues according to Tukey’s HSD test (one-way ANOVA, *p* < 0.05). (**E**) Principal component analysis based on operational taxonomic units (OTUs). *x*-axis: first principal component; *y*-axis: second principal component. The numbers after the quotation marks represent the contributions of the principal components to the differences between the samples. Dots represent individual samples, and dots with different colors represent different tissues.

**Figure 4 insects-15-00217-f004:**
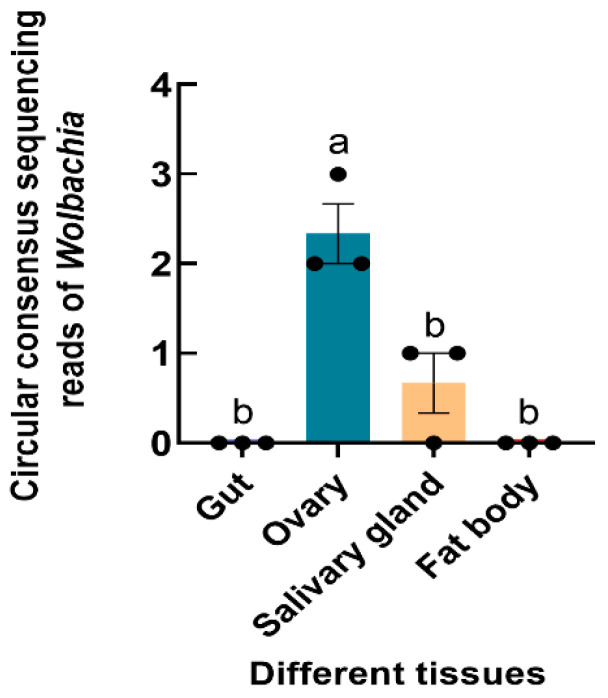
Relative abundance of *Wolbachia* in different tissues of *S. frugiperda* female adults. The columns and error bars represent the means ± SEs (n = 3). The different letters above the bars indicate significant differences between the tissues according to Tukey’s HSD test (one-way ANOVA, *p* < 0.05).

**Figure 5 insects-15-00217-f005:**
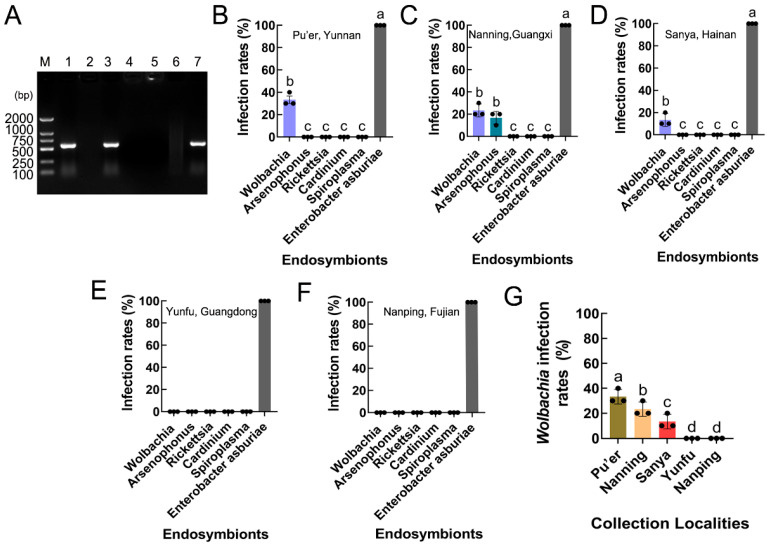
Infection and prevalence of endosymbionts in *S. frugiperda* populations. (**A**) Endosymbiont species. M: DNA marker; lane 1: positive control (*Wolbachia wsp*); lane 2: negative control (ddH_2_O); lane 3: *Wolbachia* (*wsp*); lane 4: *Spiroplasma* (*16S rDNA*); lane 5: *Cardinium* (*16S rRNA*); lane 6: *Rickettsia* (*16S rRNA*); lane 7: *Arsenophonus* (*16S rRNA*). (**B**–**F**) The rates of endosymbiont infection of *S. frugiperda* in Pu’er, Yunnan; Nanning, Guangxi; Sanya, Hainan; Yunfu, Guangdong; and Nanping, Fujian. The columns and error bars represent the means ± SEs (n = 3). The different letters above the bars indicate significant differences between the endosymbionts according to Tukey’s HSD test (one-way ANOVA, *p* < 0.05). (**G**) *Wolbachia* infection rates in different *S. frugiperda* populations. The columns and error bars represent the means ± SEs (n = 3). The different letters above the bars indicate significant differences between the collection localities according to Tukey’s HSD test (one-way ANOVA, *p* < 0.05).

**Figure 6 insects-15-00217-f006:**
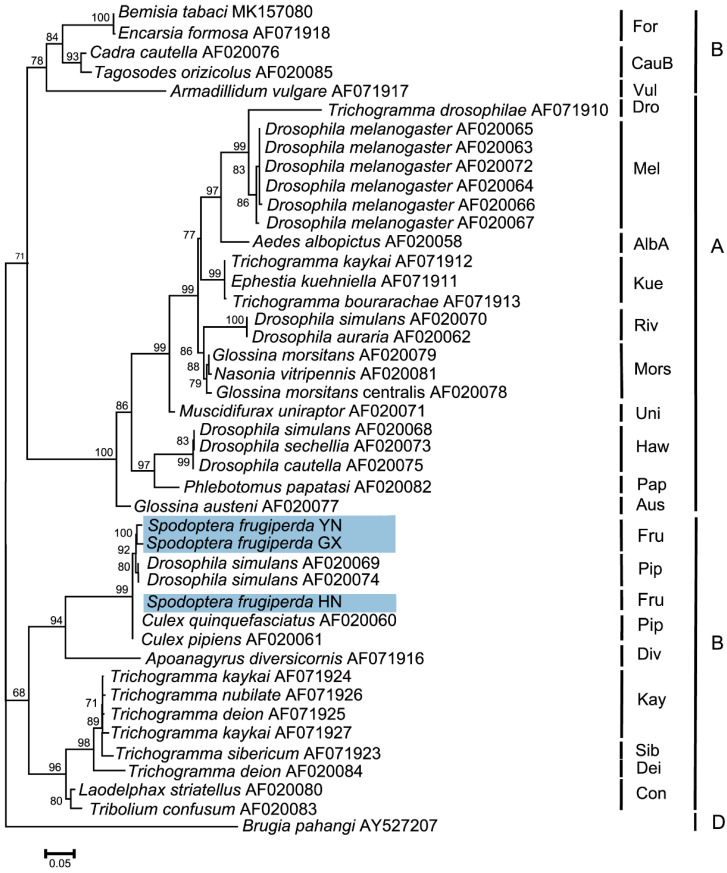
Phylogenetic analysis of *Wolbachia* based on the *wsp* gene in *S. frugiperda* populations. This phylogenetic tree was constructed and analyzed via the maximum likelihood (ML) method using 1000 bootstrap replicates. The numbers at the nodes indicate the percentages of reliability for each branch of the tree. The branch length is drawn proportionally to the estimated sequence divergence. The blue-shaded parts show the different populations of *S. frugiperda*. “YN”, “GX”, and “HN” represent Pu’er, Yunnan; Nanning, Guangxi; and Sanya, Hainan, respectively. *Brugia pahangi* was used as an outgroup.

**Table 1 insects-15-00217-t001:** Information on the *S. frugiperda* samples used in this study.

Development Stage	Number of Samples	Host	Collection Date	Collection Location	Latitude	Longitude
Adult	30	Corn	8 July 2021	Pu’er, Yunnan	22°50′38″ N	99°89′05″ E
Adult	30	Corn	5 July 2021	Nanning, Guangxi	22°60′76″ N	108°23′55″ E
Adult	30	Corn	1 December 2021	Sanya, Hainan	18°37′78″ N	109°15′16″ E
Adult	30	Corn	18 August 2021	Yunfu, Guangdong	22°72′09″ N	111°46′46″ E
Adult	30	Corn	20 August 2021	Nanping, Fujian	26°97′11″ N	117°73′25″ E

**Table 2 insects-15-00217-t002:** Details of the primers used in this study.

Target Gene	Primer Sequence (5′-3′)	Reference
*Wolbachia*	Forward: 5′-CTATAGCTGATCTGAGAGGAT-3′	[44]
(*16S rRNA*)	Reverse: 5′-YGCTTCGAGTGAAACCAATTC-3′	
*Wolbachia*	Forward: 5′-TGGTCCAATAAGTGATGAAGAAAC-3′	[45]
(*wsp*)	Reverse: 5′-AAAAATTAAACGCTACTCCA-3′	
*Arsenophonus*	Forward: 5′-CGTTTGATGAATTCATAGTCAAA-3′’	[46]
(*16S rRNA*)	Reverse: 5′-GGTCCTCCAGTTAGTGTTACCCAAC-3′	
*Rickettsia*	Forward: 5′-GCTCAGAACGAACGCTATC-3′	[47]
*(16S rRNA)*	Reverse: 5′-GAAGGAAAGCATCTCTGC-3′	
*Cardinium*	Forward: 5′-GCGGTGTAAAATGAGCGTG-3′	[48]
(*16S rRNA*)	Reverse: 5′-ACCTMTTCTTAACTCAAGCCT-3′	
*Spiroplasma*	Forward: 5′-GAGAGTTTGATCCTGGCTCAG-3′	[49]
(*16S rDNA*)	Reverse: 5′-TTCCCTTACAACAGACCTTTACAATCC-3′	
*Bacterial*	Forward: 5′-AGAGTTTGATCCTGGCTCAG-3′	[50]
(*16S rRNA*)	Reverse: 5′-GGTTACCTTGTTACGACTT-3′	

**Table 3 insects-15-00217-t003:** Samples and their processed sequence data.

Sample Name	Clean CCS Readings	Effective CCS Readings	OTU Numbers	Good’s Coverage
Gut	6892.67 ± 473.32	6881.00 ± 472.02	110.00 ± 35.38	0.9935 ± 0.0025
Ovary	5588.67 ± 398.40	5528.00 ± 427.79	194.33 ± 33.11	0.9916 ± 0.0041
Salivary gland	6981.00 ± 249.03	6828.33 ± 270.84	209.33 ± 18.80	0.9922 ± 0.0031
Fat body	7375.00 ± 83.74	7274.33 ± 71.42	44.33 ± 6.69	0.9975 ± 0.0007

Note: Data are represented by their means ± SE (*n* = 3).

**Table 4 insects-15-00217-t004:** Results of ANOSIM and PERMANOVA based on the Bray–Curtis index and weighted Unifrac distances.

Beta Diversity Distance	ANOSIM (*R*, *p*)	PERMANOVA (*R*^2^, *p*)
Bray–Curtis	0.899, 0.001	0.761, 0.001
Weighted Unifrac	0.787, 0.001	0.754, 0.001

## Data Availability

The *Wolbachia 16S rRNA* gene sequence of the *S. frugiperda* deposit GenBank was OR268559. The *Wolbachia wsp* gene sequences in the Yunnan, Guangxi, and Hainan *S. frugiperda* deposit GenBank were OR282569–OR282571.The raw sequencing data were deposited in the NCBI Sequence Read Archive (SRA) database under accession number PRJNA995535.

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
