# Peer review of "The Diversity of Wolbachia and Other Bacterial Symbionts in Spodoptera frugiperda"

_insects, 2024, doi:10.3390/insects15040217_

Round 1
Reviewer 1 Report
Comments and Suggestions for Authors
Dear Authors,
The article is interesting and reports a thorough characterization of the microbiota of S. frugiperda. However, in my opinion, several revisions are necessary to make it suitable for publication.
Language is generally fine but certain sections require a further check.
Importantly, I am not fully convinced by the interpretation of the results regarding the infection by Wolbachia and about its classification.
Also, there are various sentences that need amendment.
All my comments are referred to specific lines as follows:
Lines 15-16: Bacterial symbionts play vital roles in many physiological processes of insects, 15 among Wolbachia is one of the most abundant
The sentence is not clear
Line 75: 40- 60% of all species in nature
The Authors should specify “40-60% of all insect species”
Line 76: references 22-24 are not strictly related to the content. The authors should provide references specifically describing the presence of Wolbachia in nematodes and arthropods, highlighting the high frequency in insects.
Line 79: “to promote maternal vertical transmission”.
These modification of the hosts’ reproductive biology do not promote vertical transmission but the chance of the infected individuals to reproduce successfully and spread the infection through their progeny
Lines 79-80: “Wolbachia cannot be cultured outside host cells, and the infection rate of Wolbachia in many species of lepidopteran insects is relatively low. Consequently, understanding the composition, diversity, and potential functions of bacterial symbionts in S. frugiperda, particularly regarding Wolbachia, is still a grey area in research.”
These sentences are not clearly related.
Figure 1. the image of the ovaries is not clear enough. Ovaries should be highlighted because what I see in focus are Malpighian tubules. Anyway, the figure is only illustrative and it could be removed.
Line 171: “for 16S rRNA, wsp, and 16S rDNA genes of endosymbionts”.
The targets should be specified and more clearly associated to the primers. As an example, wsp is a gene specific to Wolbachia (encoding for the Wolbachia Surface Protein) but there are also Wsp genes (wrinkly spreader phenotype) common to Pseudomonas.
3.1. Bacterial Abundance in Different Tissues of S. frugiperda
Here, or better in introduction, the Authors should clarify the difference between endosymbiotic bacteria present inside the cells of the insect and free-living bacteria and, as a consequence, the different methods used for their recognition.
Line 262: “16S rRNA high-throughput sequencing results suggested that Wolbachia was found in the ovaries and salivary glands of S. frugiperda female adults”
how the Authors attribute Wolbachia to tissues if in methods they state that they extracted DNA from whole bodies?
Line 265-268: “The phylogenetic analysis of Wolbachia showed that S. frugiperda and Cnaphalocrocis medinalis, which are lepidopteran insects, were first clustered on one branch, which indicated that S. frugiperda and C. medinalis had closer phylogenetic relationships in the present study”.
Due to the possibility of horizontal transfer, Wolbachia is not a good method to analyze the distance between infected species. I would remove this sentence. Furthermore, a single gene is not considered enough to build phylogenetic trees regarding Wolbachia (see https://doi.org/10.3389/fmicb.2023.1084839).
Figure 6. the target genes used to detect the endosymbionts should be highlighted also here. In sub-figure G, I would use different colors to avoid confusion with figures B-F.
Lines 309-310: “Phylogenetic analysis indicated that the tested Wolbachia strains from three S. frugiperda populations belonged to the wPip strain in supergroup B”
Did the Authors find any mutation compared to the wPip Wolbachia sequence described for Culex pipiens? How they discuss the fact that Wolbachia in certain populations of S. frugiperda seem to be more closely related to the Wolbachia of D. simulans (wRi Wolbachia) instead of wPip Wolbachia?
Also in this case, using a single gene is not considered a good option to measure genetic diversity between Wolbachia strains
Lines 365-7: The phylogenetic analysis of Wolbachia showed that S. frugiperda and C. medinalis were first clustered on one branch, and had closer phylogenetic relationships.
As already discussed, I suggest to remove this sentence
Line 370-1: “Of course, the 16S rRNA sequences of Wolbachia may be extraneous, environmental contamination,”
I suggest the Author to avoid this sentence and, instead, to describe in the methods how they have prevented this possibility. Since Wolbachia detection has been conducted on wild-caught individuals and not adults emerged in the laboratory, the Authors should consider the possibility of collection of individuals carrying parasites possibly infected with Wolbachia. This hypothesis could explain why only a certain percentage of the individuals were infected and only certain populations of the species. As an example there are biting midges and mites capable to parasitize adult moths (see: https://www.floridamuseum.ufl.edu/andrei-sourakov/activities/blood-sucking-moth-parasites/). This is why at least one cycle of rearing in the laboratory is suggested before performing studies on Wolbachia. Additionally, in the case of Wolbachia infection, it is unusual that only a small proportion of the population is infected, especially in the case of CI inducing Wolbachia strains as like wPip. Wolbachia strains inducing other effects such as feminization (as commonly found in several species of moths), are instead not always fixed in the population. These considerations should guide the Authors to approach results with caution.
I encourage the Authors to provide a revised version of the article
Kind regards
Comments on the Quality of English LanguageLanguage is generally fine but it requires a further control for quality because I have noticed a few sentences that are not clear
Reviewer 2 Report
Comments and Suggestions for Authors
Basically, this study looked at the microbiome and Wolbachia infection in S. frugiperda, an important agricultural pest. These are important baseline data for this pest, and I believe they are worth publishing, subjected to some concerns listed below (may sound minor, but it's an important stance).
L 21-22: This phrase is misleading to non-experts on Wolbachia. As you know, the CI phenotype is induced by cif genes, which are often located in the prophage region of the Wolbachia genome, and proximity in phylogenetic positions does not necessarily mean that they have the same phenotype. It should be written more sincerely to avoid misunderstanding.
Figure 4 and 5: Using of only the bar graph for the data with only 3 samples can cause misunderstanding and also give the impression of insincerity. I strongly recommend adding three data points for each bar.
Comments on the Quality of English LanguageTitle: The Diversity of Bacterial Symbionts in Spodoptera frugiperda and Its Wolbachia Infection.
I think the title correctly represents what you did. Presumably, the "Its" refers to "Spodoptera frugiperda", but that is complicated and not straightforward. Why not make it as "The Diversity of Wolbachia and other Bacterial Symbionts in Spodoptera frugiperda"?
Overall, there are many problems with the English writing. Basically comprehensive, but stressful to read. I strongly recommend a thorough English editing of this manuscript.
To name a few,
L16: among -> among which
L19: comma -> that
L30: They say "five provinces", but in the following, 10 names appear. This sounds confusing for non-Chinese readers. Imagine if people in other countries do the same.
Round 2
Reviewer 1 Report
Comments and Suggestions for Authors
Dear Authors,
I thank you for taking into consideration my suggestions but, in my opinion, a few amendments are still necessary.
Summarizing, I think that it is important to avoid certain conclusions regarding the classification of the Wolbachia strain because phylogenetic trees have been obtained from the analysis of a single gene. Also, I strongly suggest to name the new strain wFru and not wPip because the Authors cannot be certain about the the idendity of this strain with that of Culex pipiens.
Further requests of minor revisions are listed below.
Lines 60-61. “The bacteria included intracellular bacteria endosymbionts and extracellular symbionts, and they differed in their residence location and interaction mechanisms with host cells [16].
The sentence is not clear and I suggest to revise it (as an example: “insects generally host a gut microbiota, that is extracellular and can vary depending on the environmental conditions and on the diet, and endosymbiotic bacteria, that are intracellular and can be obligate or facultative”.
269-272: As already written at round 1, I would eliminate this interpretation of the results because it is too speculative. On the same branch of S. frugiperda are also present Wolbachia strains from nematodes that belong to completely different Wolbachia groups. As already suggested before, using a single Wolbachia gene to build a phylogenetic tree can be misleading.
312-313. “Phylogenetic analysis of Wolbachia based on the wsp gene indicated that the tested Wolbachia strains from three S. frugiperda populations belonged to the wPip strain”
Also in this case, using a single gene is not considered appropriated to classify a Wolbachia strain (despite certain articles that are available and seemt o support thnis approach). Also based on the differences in the DNA sequence that they have shown as response to the first round of revision, I would also avoid naming it wPip (that is specific to Culex pipiens) while the Authors can reasonably attribute it to the B group. I suggest the Authors to give the strain a new name (wFru could be in accordance with the method commonly used to assign a name to a Wolbachia strain) unless they can fully sequence the genome of the Wolbachia from S. frugiperda to confirm that it is actually wPip.
Lines 372-373. “Furthermore, the 16S rRNA sequences of Wolbachia may be female adults carrying parasites, but the occurrence of this probability was relatively low and deserves our attention. The sentence is not clear”.
What do the Authors mean? The described methods seem adequate to avoid contamination by other arthropod species especially If the Authors have dissected specific organs. The sentence can be removed.
374-375. “Wolbachia is a genus of facultative endosymbionts and was reported in each of the major insect orders examined”.
The sentence can be eliminated because this information is already given in the introduction.
401-402. “In our study, phylogenetic analyses based on the wsp gene revealed that Wolbachia from different S. frugiperda populations all belonged to the wPip strain in supergroup B”
As already suggested, this conclusion is too speculative and should be removed.
402-403. “Some studies have shown that Wolbachia wPip in Culex pipiens (Diptera: Culicidae) can induce CI resulting from embryonic death [83] and block Zika virus transovarial transmission in Aedes albopictus (Diptera: Culicidae) [84]”
As a consequence of the previous comment, I suggest the Authors to only state that in Group B there are Wolbachia strains inducing CI and that this possibility should be investigated also for the Wolbachia of S. frugiperda (that they could name wFru, as suggested above)
Lines 405-406. “Two differentially genes (cifA and cifB) of prophage WO from Wolbachia wPip can induce and rescue cytoplasmic incompatibility in Anopheles gambiae [85].
It is not clear how this sentence is related to the flow of the discussion. It can be removed.
Line 411. Conclusions should also briefly point out the result regarding the other bacteria identified in this study
Comments on the Quality of English Languagelanguage quality is sufficient but a few sentences requires revision for clarity
